# The Chromatin Remodeler ATRX: Role and Mechanism in Biology and Cancer

**DOI:** 10.3390/cancers15082228

**Published:** 2023-04-10

**Authors:** Ying Pang, Xu Chen, Tongjie Ji, Meng Cheng, Rui Wang, Chunyu Zhang, Min Liu, Jing Zhang, Chunlong Zhong

**Affiliations:** 1Department of Neurosurgery, Shanghai East Hospital, School of Medicine, Tongji University, 150 Jimo Road, Shanghai 200120, China; 2Institute for Advanced Study, Tongji University, 1239 Siping Road, Shanghai 200092, China

**Keywords:** alpha-thalassemia mental retardation X-linked syndrome protein (ATRX), death-domain-associated protein (DAXX), DNA damage, replication stress, tumorigenesis

## Abstract

**Simple Summary:**

ATRX is one of the most frequently mutated tumor suppressor genes in human cancers. ATRX protein is a chromatin remodeler and transcriptional regulator that is essential for normal development. ATRX plays a crucial role in several essential cellular pathways, such as cooperating with DAXX to deposit histone variant H3.3 at repetitive regions, participating in chromatin remodeling, and responding to replication stress and DNA damage repair. ATRX mutations have been identified in several cancers and are considered important markers of clinical behavior, especially in glioma. The disruption of ATRX may contribute to cancer development and resistance to treatment. However, its role in tumorigenesis and the details of its mechanisms remain unclear. In this review, we will summarize the function of ATRX in normal biology and cancer and discuss the potential future direction of ATRX’s role in tumorigenesis. Understanding the functions of ATRX in cancers will help to develop more efficient and targeted anticancer therapies.

**Abstract:**

The alpha-thalassemia mental retardation X-linked (ATRX) syndrome protein is a chromatin remodeling protein that primarily promotes the deposit of H3.3 histone variants in the telomere area. ATRX mutations not only cause ATRX syndrome but also influence development and promote cancer. The primary molecular characteristics of ATRX, including its molecular structures and normal and malignant biological roles, are reviewed in this article. We discuss the role of ATRX in its interactions with the histone variant H3.3, chromatin remodeling, DNA damage response, replication stress, and cancers, particularly gliomas, neuroblastomas, and pancreatic neuroendocrine tumors. ATRX is implicated in several important cellular processes and serves a crucial function in regulating gene expression and genomic integrity throughout embryogenesis. However, the nature of its involvement in the growth and development of cancer remains unknown. As mechanistic and molecular investigations on ATRX disclose its essential functions in cancer, customized therapies targeting ATRX will become accessible.

## 1. Introduction

ATRX (alpha-thalassemia, mental retardation, X-linked syndrome) was identified as the gene responsible for the rare developmental condition characterized by α-thalassemia and intellectual disability [1,2,3]. ATRX is a chromatin remodeling protein that belongs to the switch/sucrose non-fermentable (SWI/SNF) family. The SWI/SNF protein family is also known as BRG1/BRM-associated factor (BAF) complexes and regulates gene expression by remodeling chromatin with ATP energy [4,5]. SWI-SNF proteins are involved in various biological activities, including DNA repair, transcription regulation, and nucleosome reorganization [6]. In recent decades, increasing evidence has demonstrated the importance of chromatin regulation in development and cancer processes. Thus, ATRX is a typical example of a gene mutation that causes cognitive disability and cancer in sequence [7].

ATRX syndrome is an X-linked severe intellectual disability characterized by mental retardation, facial dysmorphism, decreased expression of the α-globin genes (-thalassemia), urogenital dysfunction, and skeletal abnormalities [1,8]. ATRX has been considered to be an X-chromosome-encoded trans-acting factor that stimulates the expression of a chosen group of diverse genes. Subsequent investigations in several model animals have demonstrated deficiencies in a number of essential cellular processes when ATRX function is disturbed, including defective sister chromatid cohesion and congression [9,10], telomere dysfunction [11], and aberrant patterns of DNA methylation [12]. Several types of cancer, such as glioma, neuroblastoma, and pancreatic neuroendocrine tumors (PanNETs), harbor ATRX mutations [13,14,15,16,17,18,19,20,21].

The severe effects of ATRX gene mutations imply its fundamental role, which justifies the increased attention of scientists. This review provides an up-to-date summary of the various roles of the ATRX protein in both normal biology and cancer. The role of ATRX in interactions with histone variants, chromatin remodeling, DNA damage response, replication stress, and tumorigenesis will be highlighted.

## 2. Molecular Structures of ATRX

The ATRX gene is situated on the q21.1 band of the X chromosome long arm and spans almost 300 kbp. It contains 37 exons that encode a 2492 amino acid protein with a molecular weight of 282.586 kDa [22,23]. Based on its ATPase/helicase C-terminal domain, the protein is a member of the SWI/SNF family. It also has an ADD histone H3-binding cysteine-rich domain (ATRX-DNMT3-DNMT3L, ADD_ATRX_). The globular ADD domain is composed of an N-terminal GATA-like zinc finger, a plant homeodomain (PHD; Cys4-His-Cys3) finger, and a lengthy C-terminal alpha-helix (Figure 1). The exposed and very basic alpha-helix of the GATA-like finger suggests that ATRX can bind to DNA [24].

ATRX is ubiquitously expressed in the embryonic brain, suggesting a crucial involvement in brain development [23,25]. Chromatin-associated proteins that initiate and/or maintain a proper pattern of DNA methylation have the ADD domain. As a result, mutations in the ADD domain influence several cellular processes. Numerous SNF2 proteins utilize the energy released by ATP hydrolysis to translocate along DNA, therefore remodeling DNA structures or DNA-protein interactions, as well as ATRX. ATRX’s histone-binding and chromatin-remodeling capabilities play a unique role in neuronal differentiation [26]. Mutations in the ATRX protein’s PHD finger have diverse effects on the genome-wide location of polycomb repressive complex 2 (PRC2), whereas mutations in the helicase domain induce loss in some places and gains in others. Each mutation is linked to distinct gene signatures, indicating separate neurodifferentiation-impairing processes. ATRX syndrome-causing mutations commonly occur in the putative ATPase/helicase domain and the PHD motif of the ADD domain [1,2,27], while ATRX tumorigenic mutations are arbitrarily located point mutations that result in protein dysfunction (Table 1) [14,28,29].

## 3. ATRX in Regular Biology

### 3.1. Interactions with H3.3 Histone Variants

A critical role of ATRX is to deposit H3.3 in telomeres, pericentric heterochromatin, and other DNA repeat sites when it binds and interacts with the chaperone protein DAXX (death-domain-associated protein) to form a chromatin remodeling complex [30,31,32]. DAXX was initially cloned as a signaling protein that binds specifically to the transmembrane death receptor Fas (also known as CD95) and activates the JNK pathway to trigger apoptosis [33]. In the nucleus, DAXX is linked to both the promyelocytic leukemia (PML) nuclear body and ATRX-positive heterochromatic regions. In the cytoplasm, DAXX has been discovered to interact with many proteins involved in cell death regulation [34]. H3.3 is the ancestral, conserved version of H3 that is uniquely expressed outside the cell cycle and serves many functions in transcription, genomic stability, and mitosis [35]. It is feasible to substitute conventional histones with histone variants that alter the chemical components and physical characteristics of the nucleosome, thus affecting many cellular activities [36]. The HIRA complex is responsible for the deposition of H3.3 in genic regions [37,38], while the ATRX/DAXX complex is responsible for the deposition of H3.3 in repetitive sections of the genome [11,30,31].

### 3.2. Chromatin Remodeling

Chromatin is a dynamic structure that regulates the accessibility of DNA for transcription, recombination, DNA repair, and replication as well as packaging the whole eukaryotic genome within the confines of the nucleus [39]. By performing histone exchange, also known as histone turnover, cells may maintain the fluidity of chromatin. This process involves removing sections of the nucleosome or complete nucleosome, followed by replacement with freshly synthesized histones or other components. The swapping process has several effects on the composition, structure, and function of various genomic areas. During transcription, several factors regulate histone exchange. In eukaryotes, chromatin-remodeling complexes play an important role in gene expression regulation [40,41]. These complexes can alter the structure of the chromatin in two different ways. The first is covalent modification, which includes methylation, phosphorylation, and acetylation. Non-covalent processes, such as ATP-dependent chromatin remodeling, are involved in the other.

ATRX interacts directly with DNA and collaborates with many functional partners to regulate the structure and function of chromatin in centromeric heterochromatin and telomeric domains. The accessibility of chromatin (both in repetitive and non-repetitive DNA) varies significantly in the absence of ATRX, resulting in the transcription of normally repressed areas [42,43]. ATRX loss produces chromatin decompaction at telomeres and repetitive elements [44,45,46]. ATRX deficiency causes severe chromosomal cohesion and congression abnormalities in Hela cells [10]. ATRX regulates the expression of certain imprinted genes in the brain by modifying the formation of chromatin loops and large-scale chromatin structures [47]. Except for its function in H3.3 deposition, ATRX also collaborates with H3.3 and chromobox homolog 5 (CBX5) to maintain telomeric RNA transcriptional suppression in mouse and human ES cells [11,30]. Essentially, ATRX loss results in chromatin modifications and telomere instability in mouse ES cells [11].

ATRX is required for chromosomal stability throughout both mitosis and meiosis. ATRX is necessary for centromere stabilization and epigenetic regulation of heterochromatin function throughout meiosis and the transition to the first mitosis [48]. During prophase I arrest, ATRX is necessary to bind the transcriptional regulator DAXX to pericentric heterochromatin. At the metaphase II stage, transgenic ATRX-RNAi oocytes have an aberrant chromosomal shape linked to decreased histone 3 phosphorylation in serine 10, and chromosome segregation problems that result in aneuploidy and drastically diminished fertility. ATRX also regulates crucial phases of meiosis in mouse oocytes [9]. Global histone deacetylation at the beginning of meiosis, for example, is required for ATRX binding to centromeric heterochromatin in mouse chromosomes. Centromeric ATRX is also necessary for proper chromosomal alignment and spindle organization inside the bipolar meiotic metaphase II spindle.

The ATRX-DAXX complex was reported to be a new ATP-dependent chromatin-remodeling complex, with ATRX as the main ATPase component and DAXX as the targeting subunit [49]. ATRX and its transcription cofactor DAXX were identified as a complex by immunoprecipitation from HeLa extract, and the complex exhibits ATP-dependent activities similar to those of other chromatin-remodeling complexes, such as DNA displacement and modification of mononucleosome disruption patterns [50]. DAXX attaches directly to H3.3 because its hydrophobic pocket is shallow enough to accept tiny hydrophobic Ala87 of H3.3, and its polar binding environment favors hydrophobic Gly90 in H3.3 over hydrophobic Met90 in H3.1 [51]. The ATRX-DAXX complex recognizes H3K9me3 via the ADD domain and is likely responsible for recruiting its binding partner DAXX into these areas for H3.3 deposition [52,53,54,55] (Figure 2).

### 3.3. DNA Damage Response

The DNA damage response (DDR) is a complex network of signaling pathways that cells establish to respond to exogenous or endogenous DNA damage that triggers genetic changes. ATRX works during DNA replication and is directly involved in DNA repair. ATRX deficiency increases sensitivity to agents, which induces replication stress [56]. Recent investigations have demonstrated that ATRX protects hydroxyurea (HU)-stalled replication forks and promotes their restarting [56,57,58]. ATRX is essential for proficient S phase progression by limiting fork stalling by recruiting to DNA damage sites and interacting with components of the MRE11-RAD50-NBS1 (MRN) complex, which has several known functions crucial for genomic stability and replication, such as the repair of double-strand breaks and the restart of a stalled replication fork [56,57].

ATRX is a unique functional partner of FANCD2 in the S phase that promotes histone deposition-dependent homologous recombination (HR) processes, according to recent studies [59]. FANCD2 is the protein of the central FA (Fanconi anemia) pathway that recruits HR factors such as the CtBP interacting protein (CtIP) to enhance replication fork restart while inhibiting new origin firing [60,61]. ATRX and FANCD2 build a complex that prevents proteasomal degradation of FANCD2. ATRX further collaborates with FANCD2 to recruit CtIP and promote meiotic recombination 11 (MRE11) exonuclease-dependent fork restart, while inhibiting the firing of new replication origins. ATRX and FANCD2 interact to facilitate HR-dependent repair of directly generated double-strand breaks in DNA (Figure 3).

ATRX promotes DNA repair synthesis and sister chromatid exchange HR [62]. DNA double-strand breaks (DSBs) occur when cells are exposed to endogenous or exogenous stress. DSBs can be repaired by two major pathways: non-homologous end-joining (NHEJ) and HR [63]. HR is initiated by long-range 5′ end resection and RAD51 loading onto single-stranded DNA. Later phases comprise homology search, invasion of DNA strips to create a displacement loop (D loop), removal of RAD51, and repair synthesis to duplicate missing sequence information from a donor sister chromatid at the location of the break [64]. ATRX-deficient cells cannot repair exogenously induced DSBs by HR. ATRX and DAXX deposit the histone variant H3.3 during HR-mediated repair of exogenously induced DSBs. ATRX functions after the removal of RAD51 and interacts with PCNA and RFC-1, all of which are essential for DNA repair synthesis during HR. Following a RAD51-dependent homology search, the deposition of H3.3 promotes extended DNA repair synthesis and the establishment of sister chromatid exchanges. Therefore, ATRX enables the reconstitution of chromatin necessary for extended DNA repair synthesis and sister chromatid exchange during HR (Figure 4).

ATRX participates in the DNA damage repair pathway by modulating the ATM pathway [65]. Ataxia telangiectasia mutated (ATM) kinase is an essential enzyme that detects and repairs DSBs caused by temozolomide (TMZ) or radiation [66,67]. ATM activation is regulated by several variables, such as TIP60 acetyltransferase activity and histone H3K9me3 status [68,69]. ATM phosphorylation is mediated by its acetylation level, whereas histone H3K9me3 is necessary for ATM acetylation in response to DNA damage [70]. ATRX can build a complex to promote the deposition and maintenance of H3K9me3 [71]. ATRX is recruited to pericentromeric heterochromatin by the interaction between H3K9me3 and its ADD domain. Mutations in this domain prevent ATRX from binding to H3K9me3, which may result in chromosome mis-segregation and apoptosis in neuroprogenitor cells [53]. ATRX deletion decreased the association between H3K9me3 and ATRX, which may inhibit TMZ-induced ATM acetylation. ATRX deficiency promotes the H3K9 trimethylation status that prevents ATM phosphorylation and leads to the deactivation of the ATM pathway.

### 3.4. Response to Replication Stress

ATRX is hypothesized to promote genome stability by avoiding replication stress via the resolution of G-quadruplex (G4) DNA structures ahead of the replication fork [72]. When double-stranded DNA is detached in areas rich in GC during replication and transcription, these stable non-B-form DNA structures are predicted to arise [73,74]. The G4 structures are believed to improve replication stress by inhibiting DNA replication fork advancement, resulting in replication fork collapse and DNA breakage [75]. ATRX is bound in GC-rich regions with a proclivity for forming G4 structures throughout the genome, and ATRX directly interacts with G4 structures in cells [72,76]. ATRX presumably assists in the replication of telomeric G4-DNA structures [77], its absence results in the aggregation of G4 structures at DNA synthesis sites [72], and exogenous expression of ATRX in ATRX defect cells could decrease the levels of G4 structures [78]. ATRX has been demonstrated to protect cells from replication stress caused by CX-5461 (or CX-3543), the chemical stabilizer of G4 structures [78,79]. Furthermore, the function requires ATRX helicase activity and ATRX/DAXX-mediated H3.3 deposition, but not HIRA-mediated H3.3 deposition [72]. ATRX interacts with DAXX by promoting the deposition of the histone variant H3.3 to sustain G4-containing areas in a closed heterochromatic state. The formation of heterochromatin is the critical biochemical step that protects cells from G4-mediated replication stress. However, the detailed molecular process of ATRX recruitment and function in G4 areas remains unclear and requires future research.

## 4. ATRX and Cancer

ATRX has been reported as a tumor suppressor that is frequently mutated in a variety of tumors, including adult lower-grade gliomas, pediatric glioblastoma multiforme, pediatric osteosarcoma, neuroblastoma, and pancreatic neuroendocrine tumors [13,14,19,21]. Acquired somatic mutations in ATRX were primarily identified in patients with the rare subtype of myelodysplastic syndrome (MDS) associated with thalassemia (ATMDS) [80,81]. Mutations of ATRX include point mutations in the coding regions and deletion/insertion-induced frameshift mutations that lead to functional loss [82]. Recent finding indicates that mutations in ATRX are associated with a specific subgroup of tumors that are characterized by alternative telomere lengthening (ALT), an aberrant telomerase-independent form of HR-based telomere maintenance. A new study found that ATRX loss and mutations are hallmarks of 90% ALT-immortalized cell lines [83]. Moreover, ATRX mutations appear to be mutually exclusive to mutations in the promoter of the telomerase reverse transcriptase (TERT) gene that increase telomerase expression [84]. These studies strongly suggest that ATRX is a suppressor of ALT and plays an important role in tumorigenesis. However, loss-of-function mutations of ATRX alone are not sufficient to drive the ALT process. ALT activation is a multifactorial, cell-type-specific process, with ATRX/DAXX mutations being just one contributing factor [85,86,87,88].

ATRX is one of the twenty most frequently mutated genes in cancer and is the third most mutated gene in gliomas, as registered in the National Cancer Institute GDC data portal (accessed on 25 March 2023). More and more evidence shows that ATRX is implicated in cancer initiation, progression, therapy, and therapeutic resistance. Here, we provide a summary of its function in gliomas, neuroblastomas, and pancreatic neuroendocrine tumors (Figure 5).

### 4.1. ATRX in Gliomas

Infiltrating gliomas are the most common primary malignant brain tumors, accounting for 75% of primary malignant brain tumors in adults, which are typically associated with a poor prognosis and low quality of life [89]. Gliomas derive from glial or precursor cells and consist of astrocytomas, oligodendrogliomas, and ependymomas [90]. Recurrent mutations in the ATRX gene are associated with an alternate telomere lengthening phenotype [15,91]. In pediatric glioblastoma (GBM), somatic mutations in the H3.3-ATRX-DAXX chromatin remodeling pathway were reported in 44% of tumors. Furthermore, ATRX mutations were identified in 31% of individuals with primary GBM (WHO grade IV glioma) [16]. In adults, ATRX mutations are less prevalent in primary GBM, but more common in lower grade (WHO grade II or III) and secondary glioblastomas [92,93]. ATRX mutations are important markers of clinical behavior, and are strongly associated with IDH (isocitrate dehydrogenase) mutations, closely correlated with *TP53* mutations, and mutually exclusive with 1p/19q codeletion [13,17]. In the 2016 World Health Organization classification of Central Nervous System Tumors, the ATRX status was incorporated into the diagnostic algorithm for glioma variants combined with histology (Table 2) [94,95].

ATRX mutations in gliomas are associated with a better prognosis and longer patient survival [82,96]. The function of ATRX mutations in human gliomas has been revealed. Recent research provides insight into the significance of ATRX mutations in human gliomas [97]. In this paper, the scientists established an animal model of ATRX-deficient GBM utilizing the Sleeping Beauty (SB) transposase system. They discovered that the absence of ATRX impaired glioma tumor proliferation and led to genetic instability, including microsatellite instability and telomere maintenance impairment. In the investigation of publicly available human glioma genome-wide data, ATRX mutations were related to a higher mutation rate at the single-nucleotide variant level, but not at the chromosomal/copy number level. They also indicated that ATRX deficiency impairs NHEJ, which is significantly associated with the loss of active (phospho-) DNA-dependent protein kinase catalytic subunit (pDNA-PKcs) staining, thus increasing the sensitivity to DNA-damaging chemicals that induce double-stranded DNA breaks. Their investigation provides a mechanism for genetic instability and an actionable therapeutic target for ATRX-deficient GBM.

The function of ATRX in DNA replication and repair has been increasingly emphasized. Loss of ATRX inhibits ATM-dependent DNA damage repair by regulating H3K9me3 modification to increase TMZ sensitivity in gliomas [65]. Previous studies found a link between ATRX expression and the level of DNA methylation of chromosome ends in gliomas [98,99]. In the current work, ATRX knockout glioma cell lines were established by CRISPR/Cas, and impaired proliferation and migration, as well as improved sensitivity to TMZ, were observed. In addition, they verified a decreased activation of the ATM pathway mediated by the H3K9 trimethylation status. By modulating the ATM pathway, these data show that ATRX is important in DNA damage repair.

ATRX-deficient GBM cells are not only more sensitive to TMZ but also to irradiation [97,100,101]. To explain the proliferative alterations and responsiveness to irradiation observed in ATRX mutant human gliomas, researchers discovered that ATRX binds to regulatory elements of genes involved in the cell cycle phase transition in murine neuronal progenitor cells (mNPCs) and mGBM neurospheres [102]. Checkpoint Kinase 1 (CHEK1), the essential cell cycle checkpoint regulating gene, was significantly down-regulated by ATRX deficiency in numerous high-grade glioma (HGG) models, resulting in the early release of G2/M entry following irradiation. Further results showed that in response to irradiation and ATM inhibition-targeted sensitization, ATRX-deficient GBM cells exhibited a decreased capacity to maintain the G2/M cell cycle checkpoint. Consequently, the combination of irradiation and ATM inhibitors provides a novel synthetic lethal therapy for ATRX-deficient glioma (Figure 6).

ATRX also contributes to TMZ resistance in gliomas [103]. GBM has a distinct anti-DNA damage phenotype that is responsible for chemoresistance [104]. Even though TMZ is the first-line treatment for GBM, drug resistance is a major problem in therapy [105]. In gliomas, genetic inactivation of ATRX was found to impair cell proliferation and increase TMZ-induced DNA damage [65]. Further evidence showed that ATRX expression was increased by DNA demethylation caused by the STAT5b/TET2 complex in TMZ-resistant glioma cells [103]. PARP1, a member of the poly (ADP-ribose) polymerase family, plays an essential role in DSBs repair [106]. ATRX increased PARP1 stability by inhibiting H3K27me3 enrichment in the Fas-associated death domain (FADD), a necroptosis factor that regulates PARP1 cleavage [107]. The loss of ATRX confers sensitivity to PARP inhibitors, which has been associated with increased replication stress [108]. In TMZ-resistant xenograft animal models, the combination of PARP inhibitors with TMZ decreased glioma development, suggesting the possibility of a synthetic lethal strategy for overcoming ATRX-mediated TMZ resistance in gliomas.

Loss of ATRX suppresses anti-tumor immunity [109]. IDH 1/2 mutations characterize a subtype of glioma with a better prognosis and unique ontogeny than IDH-wildtype glioma [110]. According to the 2016 WHO standard, Grade II/III IDH-mutant gliomas are divided into oligodendrogliomas (IDH-O) and astrocytomas (IDH-A) according to the presence of 1p/19q co-deletion or 1p/19q-intact [95]. The loss of function in ATRX is a characteristic of IDH-A, while ATRX mutations are infrequent in IDH-O. Recent research has found that ATRX regulates the tumor microenvironment in IDH-mutant gliomas [111]. The single-cell transposase-accessible chromatin (scATAC-seq) and sc/snRNA-seq data from 22 untreated IDH-A/O human gliomas reveal cell-type-specific differences in transcription-factor use, related targeting, and cis-regulatory grammars between IDH-A and IDH-O. The proliferation of IDH-A cells is facilitated by nuclear factor I (NFI) transcription factors; they up-regulate the nuclear factor kappa-light-chain-enhancer of activated B cells (NFκB) pathway genes and subsequent cytokine production. ATRX deficiency causes global loss of CCCTC-binding factor (CTCF, which is localized by H3.3 histones) and boundary disturbances of the chromatin loop, which promote coordinated loop-wide increases or decreases in gene expression, resulting in the observed phenotypes. ATRX deficiency in IDH-mutant gliomas orchestrates chromatin and gene-expression variations that govern glial identity and myeloid-cell induction. These findings are consistent with previous research showing that ATRX loss increases astrocytoma cell aggressiveness by inducing immunosuppressive gene expression in IDH-mutant gliomas [112]. Loss of ATRX upregulated the immune-checkpoint protein programmed death-ligand 1 (PD-L1) and the production of immunosuppressive cytokines (e.g., IL33, CXCL8, CSF2, IL6, CXCL9). Furthermore, the absence of ATRX enhanced tumor cells’ resistance to T-cell killing and induced T-cell apoptosis, tumorigenic/anti-inflammatory macrophage polarization, and Treg infiltration. In addition, chemoradiation amplified the impact of ATRX loss on immune modulator expression. The transcriptional and biological immune-suppressive responses to ATRX loss depend on the expression of bromo- and extra-terminal (BET, epigenetic readers of acetylated lysine residues) proteins BRD3/4, which were abrogated by pharmacologic BET inhibition. These studies provide novel therapeutic strategies and require a comprehensive comprehension of the reprogramming of the tumor microenvironment induced by ATRX inactivation.

ATRX has an important clinical significance for gliomas. ATRX loss is a useful biomarker in improving the diagnosis of IDH mutant astrocytomas and may be used to delineate these tumors from oligoastrocytomas and oligodendrogliomas [113,114]. ATRX and IDH mutant anaplastic astrocytomas have a favorable prognosis than anaplastic astrocytomas with only IDH mutation [82]. Recently, large-scale studies of more than 400 oligodendroglial and astrocytic gliomas have further strengthened the notion that ATRX is an important diagnostic marker [113,114,115]. Unlike pediatric GBM that ATRX mutations occur at a hotspot near the carboxyl terminal helicase domain, adult glioma mutations are evenly distributed in all genes. ATRX deletion occurs almost exclusively in IDH mutation tumors, and ATRX deletion and 1p/19q codeletion are largely mutually exclusive [13,82]. Exome sequencing of IDH mutations, 1p/19q intact and oligoastrocytoma showed a high incidence of mutations in the ATRX gene, but ATRX deletion rarely occurs in 1p/19q codeleted oligodendroglioma [113].

### 4.2. ATRX in Neuroblastomas

Neuroblastoma is a common and aggressive pediatric neuronal tumor that emerges from the developing sympathetic nervous system and has poor overall survival [116,117]. In whole genome sequencing analyses of high-risk neuroblastoma, MYCN amplifications (37%), TERT rearrangements (23%), and recurrent ATRX deletions (11%) were identified [18,118,119]. The amplification of MYCN and the age at diagnosis are the two significant predictors of the outcome, with the outcomes gradually worsening with increasing age at diagnosis [116,120]. ATRX alterations are prevalent in neuroblastoma in adolescents and young adults, which is linked with overall poor survival and the lack of effective treatments [121,122].

ATRX loss-of-function mutations are a potential cause of pediatric cancer biology. ATRX mutations have identified a subtype of neuroblastoma with a different clinical phenotype, including an older age at diagnosis, resistance to traditional therapy, and a chronic but progressive disease course [121]. To evaluate the impact of loss of ATRX function in neuroblastoma, Sally L George et al. established neuroblastoma cell lines isogenic for ATRX by CRISPR-Cas9 gene editing [123]. They found that ATRX deficiency led to impaired DNA damage repair through HR and impaired replication fork processivity. This is consistent with the high-throughput drug screening findings that ATRX mutant cells are selectively sensitive to various PARP inhibitors and the ATM inhibitor KU60019. Then, the combination of the PARP inhibitor Olaparib with the DNA-damaging agent irinotecan is effective in preclinical neuroblastoma models with genetic alterations in ATRX. ATRX deficiency leads to particular DNA damage repair defects that can be therapeutically exploited.

Amplification of the MYCN oncogene and inactivation of the ATRX tumor-suppressor gene are associated with high-risk disease and poor prognosis in neuroblastoma. However, ATRX mutations and MYCN amplification are incompatible in neuroblastoma of all ages and stages [124]. MYCN, a member of the larger MYC family, regulates various cellular processes during development and in cancer. As a proto-oncogene, MYCN is frequently deregulated in human cancers, and MYC-dependent metabolic reprogramming is critical for tumorigenesis [125,126]. Increased MYCN levels induce metabolic reprogramming, mitochondrial dysfunction, the production of reactive oxygen species, and DNA replication stress. One critical function of ATRX is to protect cells from G4-mediated replication stress, which can block DNA replication or transcription, resulting in replication fork collapse [127,128]. ATRX mutations in neuroblastoma increase replication stress and cause DNA damage repair defects. Consequently, the DNA-replicative stress caused by ATRX mutations and MYCN amplification causes synthetic lethality in neuroblastoma, representing an uncommon instance in which the inactivation of a tumor-suppressor gene and activation of an oncogene are incompatible.

In addition to the detected point mutations and indels at the ATRX site, the large N-terminal deletions of ATRX generate in-frame fusion proteins (IFF), which lack several important chromatin interaction domains and contribute to aggressive neuroblastoma through reorganization of the chromatin landscape and result in transcriptional dysregulation. Recent research has demonstrated that ATRX in-frame fusion neuroblastomas are sensitive to the enhancer of zeste homolog 2 (EZH2) inhibition by modulating neuronal gene signatures [129]. ATRX has a helicase domain similar to SWI/SNF that modulates DNA accessibility [27]. The ATRX IFFs found in neuroblastoma lack most of these chromatin-binding modules, leading to alterations in their genomic binding. The ATRX IFF proteins are reallocated from H3K9me3-enriched chromatin to promoters of active genes, notably the RE-1 silencing transcription factor (REST). REST is a transcriptional repressor that binds to RE1 motifs, which are neuron-restrictive silencer elements [130,131]. The main function of REST is to suppress neuronal gene transcription in non-neuronal cells, which is important in neuronal development [132]. The research identifies REST as an ATRX IFF target whose activation increases the silence of genes involved in neuronal differentiation. Further evidence showed that REST deficiency and EZH2 inhibition enhance neuronal genes derepression and cell death (Figure 7). These findings support the notion that therapeutic targeting of ATRX IFF neuroblastoma with EZH2 inhibitors is a potential therapy for this aggressive neuroblastoma subtype.

The frequency of ATRX mutations in neuroblastoma was substantially higher in older patients, i.e., children older than five years, adolescents, and young adults. Although neuroblastoma is rare in older individuals, the mutations in ATRX seem to have significant predictions, because individuals with somatic mutations seem to have a chronic but progressive and lethal disease course [121]. Neuroblastoma tumors with ATRX mutations had lengthened telomeres, and anti-telomerase-based therapies might be valuable [133]. ATRX-deficient neuroblastoma cells exhibit increased sensitivity to the ATM inhibitor KU60019 [123]. The understanding of ATRX molecular functions will provide discoveries of potential cancer treatments.

### 4.3. ATRX in Pancreatic Neuroendocrine Tumors

Pancreatic neuroendocrine tumors (PanNETs) are rare and genetically heterogeneous; they account for around 3% of all pancreatic tumors and have a high malignant potential [134]. More than 50% of patients will die from their tumor within 10 years, as there are no effective therapies other than surgery. PanNETs may be functional or non-functional based on hormonal symptoms, with the latter being more common. These hormones include insulin, gastrin, glucagon, vasoactive intestinal peptide, and somatostatin. Functional PanNETs present at an early stage due to tumor-related symptoms and complications, non-functional PanNETs are often diagnosed at a later stage, when the illness has progressed locally or metastasized [135]. Despite improvements in prognostic grading and staging systems, the prediction of clinical behavior and response to specific therapies remains a challenge. DNA methylation is essential for tumorigenesis and could contribute to the identification of PanNET subgroups, and these subgroups could potentially be associated with clinical features [136,137,138]. A deeper understanding of the molecular mechanisms leading to the development of PanNETs is needed.

Chromosomal instability (CIN) is a characteristic of malignant PanNETs that was detected in patients with poor outcomes [139]. Whole-exome sequencing recently revealed very frequent somatic mutations in DAXX and ATRX in PanNETs, which were found to be beneficial [14,140]. Both DAXX and ATRX mutations in PanNETs are associated with ALT activation [28]. The absence of the ATRX, DAXX, and subsequent ALT phenotype activation are related to CIN in PanNETs and are associated with increased metastatic potential in sporadic primary PanNETs [141,142,143]. Therefore, it was considered a biomarker but has not been included in clinical decision making to date [144,145]. A recent study has indicated that the loss of chromatin-remodeling proteins and/or cyclin dependent kinase inhibitor 2A (CDKN2A) is linked to PanNET metastasis and shorter patient survival times [146]. Set domain containing 2 (SETD2) protein is a histone H3 lysine trimethyltransferase, and loss-of-function mutations lead to the absence of H3K36me3 expression [147]. Defects in ARID1A (AT-rich interaction domain 1A), a component of the SWI/SNF chromatin remodeling complex, increase the sensitivity of tumor cells to ATR inhibitors [148]. CDKN2A is a tumor suppressor and a cyclin-dependent kinase inhibitor which is essential in cancer [149]. The authors found that loss or deletion of either DAXX, ATRX, H3K36me3/SETD2, ARID1A, or CDKN2A in primary PanNETs leads to significantly shorter patient survival rates.

The absence of ATRX/DAXX is frequent in PanNETs [140], indicating that the complex might play a crucial role in the pathogenesis. Moreover, loss of ATRX/DAXX expression is a late event in pathogenesis that is associated with an aggressive phenotype [150]. To explore the exact mechanisms of how ATRX and DAXX mutations make sense of tumorigenesis, relevant preclinical experimental models are required. Genetically engineered mouse models are excellent tools for investigating the multistep tumorigenic pathway of PanNETs and assessing the role of candidate genes in tumor initiation and progression [151]. The group of Amanda R. Wasylishen used genetically engineered mouse models combined with environmental stress to evaluate the tumor suppressor functions of DAXX and ATRX in the mouse pancreas [152]. They found that DAXX or ATRX loss, alone or in combination with MEN1 loss, did not drive or accelerate pancreatic neuroendocrine tumorigenesis. The results strongly show that the human genome is crucial to promote tumor growth after the loss of ATRX or DAXX. Another group developed a genetically engineered mouse model with ATRX conditional disruption in β cells to investigate the potential role of ATRX as a driver event in PanNET tumorigenesis [153]. However, they verified that ATRX deficiency did not cause PanNET formation but led to dysglycaemia and the exacerbation of inflammageing (increased pancreatic inflammation and hepatic steatosis).

Mutations in ATRX are found in about a third of sporadic non-functional PanNETs and are accompanied by DAXX mutations and ALT [140]. In all neuroendocrine neoplasms, ATRX/DAXX mutations and ALT are found almost exclusively in PanNETs and are not present in other cellular origins [154]. The absence of ATRX increases CIN and mutational burden. However, ATRX/DAXX protein loss is not the initiating genomic alteration but rather occurs at a later stage in the development of the primary NF-PanNET that is associated with the progression to metastatic disease [145]. Therefore, loss of ATRX/DAXX protein and ALT in primary PanNET is a strong prognostic biomarker of recurrence and/or development of metachronous metastatic disease [154,155,156]. In numerous retrospective studies, ALT and/or ATRX/DAXX protein loss is the strongest predictor of recurrent disease after surgery.

## 5. Conclusions

Since the ATRX gene was found, various functions of ATRX that are involved in many essential cellular pathways have been identified. ATRX functions as a chromatin-remodeling complex together with DAXX to deposit histone variant H3.3 at repetitive regions [52]. ATRX plays a crucial role the in the dynamic regulation of chromatin structure and the responses to replication stress and DNA damage repair (Figure 8). ATRX plays an essential role in chromatin remodeling, especially in the resolution of G4 DNA structures, but the detail of the molecular mechanism needs more research to be revealed. Although ATRX participates in DNA repair pathways such as HR and NHEJ, it remains unclear if ATRX promotes DNA repair through direct involvement in specific pathways or plays a role in controlling the balance or activity of different repair pathways. The disruption of ATRX has been related to several cancer alterations that contribute to cancer development and progression or resistance to treatment. In regard to the high frequency of ATRX mutations in cancer, the chromatin regulator appears to play a key role in pathogenesis. However, the details of how ATRX regulates cell fate decisions during development that go awry in cancer remain unclear. Given the diversity of mutations identified in ATRX, there remains a question of whether all mutations are loss-of-function, such as in-frame deletions of ATRX in neuroblastoma [129]. As the complete absence of ATRX is not tolerated in development, we must ask whether ATRX alterations are truly loss-of-function or if they are hypomorphic. Considering that ATRX is related to immunological responses in cancer, molecular studies focusing on this area will offer an opportunity to promote immunotherapy. This paper provides an overview of ATRX, including both structure and functions. More research is needed to investigate the role of ATRX in tumorigenesis and to reveal new therapeutic approaches.

## Figures and Tables

**Figure 1 cancers-15-02228-f001:**
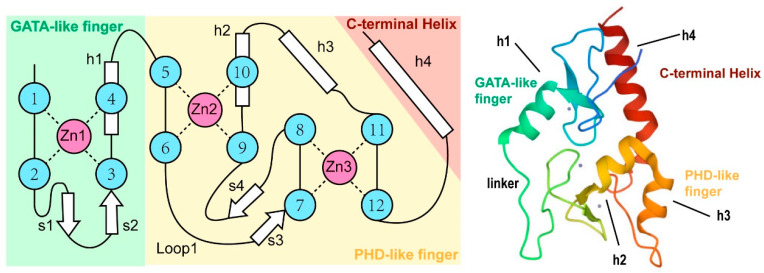
Schematic showing the zinc-binding topology and secondary structure elements of the ADD domain. Β-Strands are labeled s1–s4 and helices are labeled h1–h4. Ribbon representation of the NMR structure of the ADD domain of ATRX.

**Figure 2 cancers-15-02228-f002:**
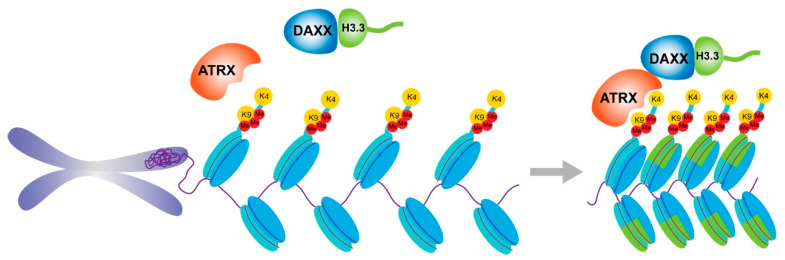
ATRX recognized H3K9me3 and worked with DAXX to deposit H3.3 at telomeres and preserve heterochromatin.

**Figure 3 cancers-15-02228-f003:**
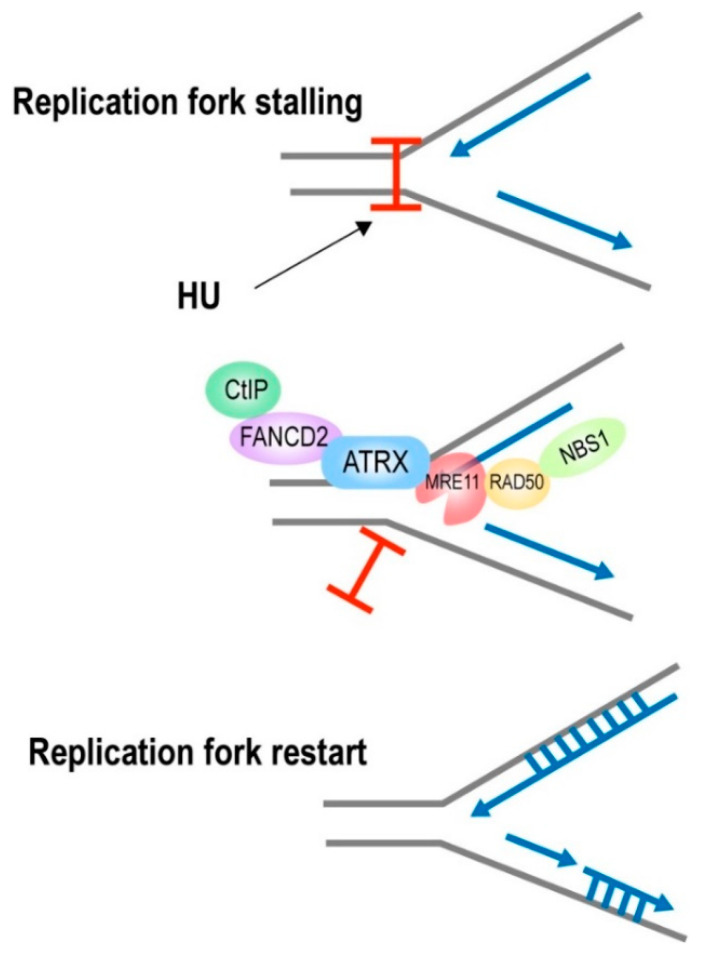
ATRX cooperated with FANCD2 and MRE11 to respond to the HU-induced replication stress and promote replication restart. Abbreviations: MRE11, meiotic recombination 11; HU, hydroxyurea.

**Figure 4 cancers-15-02228-f004:**
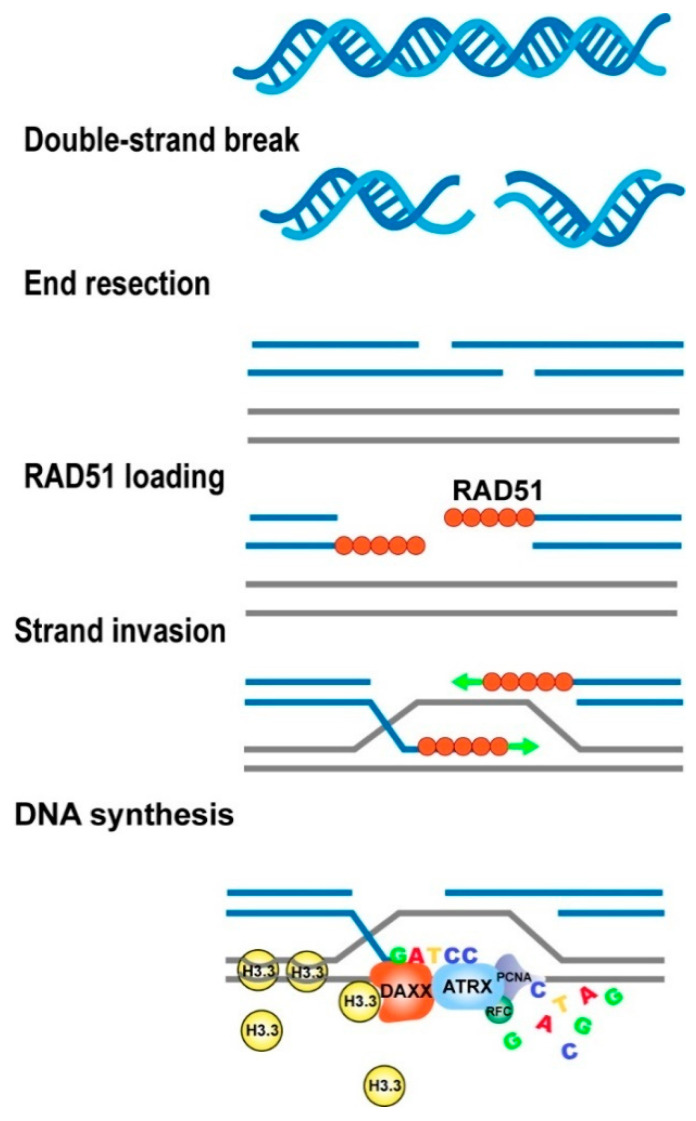
The ATRX-DAXX complex deposits the histone variant H3.3 during HR-mediated repair of DSBs. Abbreviations: HR, homologous recombination; DSBs, double-strand breaks.

**Figure 5 cancers-15-02228-f005:**
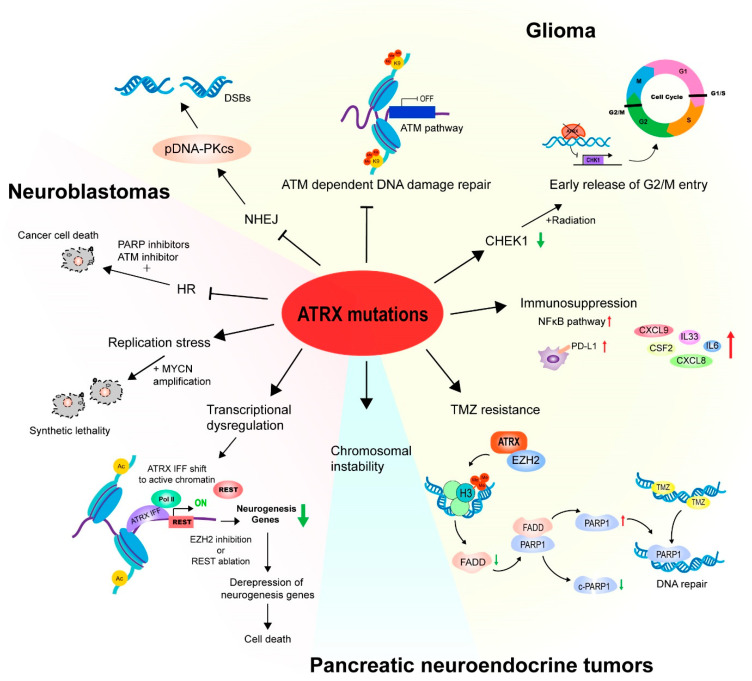
ATRX mutations and cancers. In gliomas, ATRX mutations impair NHEJ and ATM-dependent DNA damage repair pathway, down-regulate CHEK1 expression that results in the early release of G2/M entry, induce immunosuppression, and participate in TMZ resistance. In neuroblastomas, ATRX mutations impaired DNA damage repair through HR, cause synthetic lethality with MYCN amplification, and regulate transcription through reorganization of the chromatin landscape. In pancreatic neuroendocrine tumors, ATRX mutations lead to chromosomal instability. Red arrows: up-regulate; green arrows: down-regulate. Abbreviations: HR, homologous recombination; NHEJ, non-homologous end-joining; PARP, poly (ADP-ribose) polymerase; CHEK1, Checkpoint Kinase 1; PD-L1, immune-checkpoint protein programmed death-ligand 1; TMZ, temozolomide; EZH2, enhancer of zeste homolog 2; FADD, Fas-associated death domain; ATRX IFF, ATRX in-frame fusion proteins.

**Figure 6 cancers-15-02228-f006:**
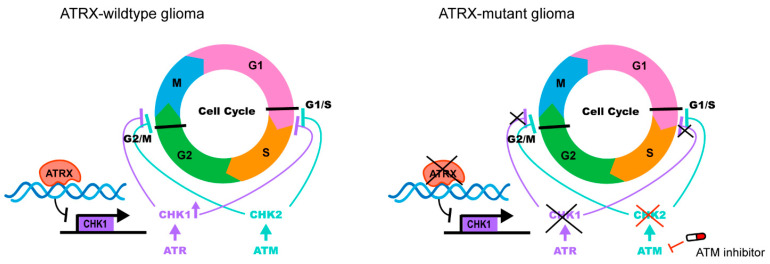
ATRX binds the regulatory elements of the cell cycle phase transition gene CHEK1 in glioma. ATRX deficiency results in reduced CHK1, which increases reliance on ATM. ATRX loss impaired the ability to maintain the G2/M cell-cycle checkpoint after radiation, and improved radio-sensitization with ATM inhibitors.

**Figure 7 cancers-15-02228-f007:**
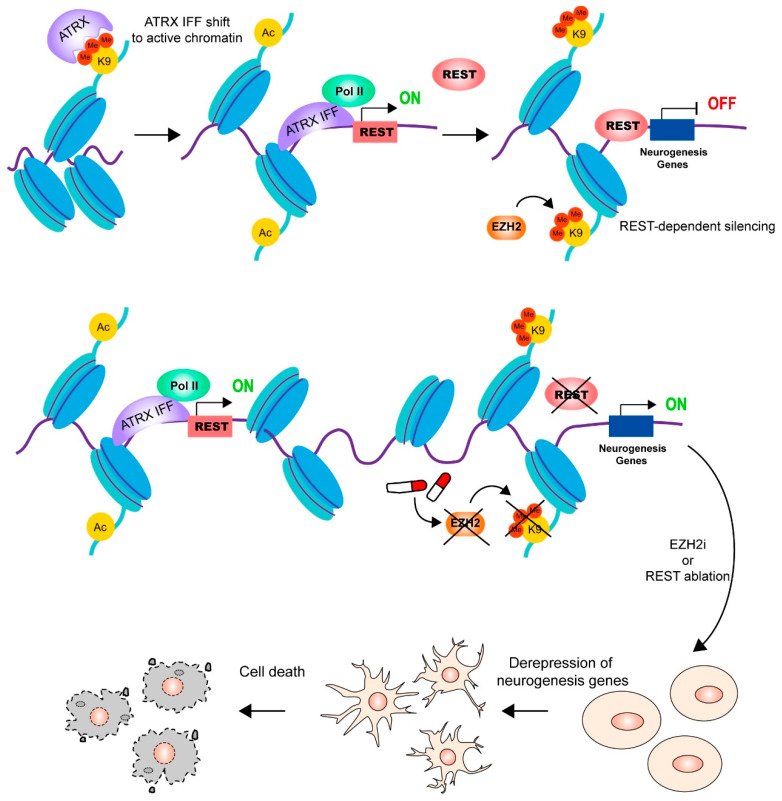
ATRX IFFs localize from H3K9me3 to the REST promoter, leading to its activation. REST and EZH2 cooperate to silence neurogenesis genes. REST depletion or EZH2 inhibition upregulates neurogenesis genes, which prompt differentiation followed by apoptosis. Abbreviations: ATRX IFF, ATRX in-frame fusion proteins; REST, RE-1 silencing transcription factor; EZH2, enhancer of zeste homolog 2; EZH2i, EZH2 inhibitor.

**Figure 8 cancers-15-02228-f008:**
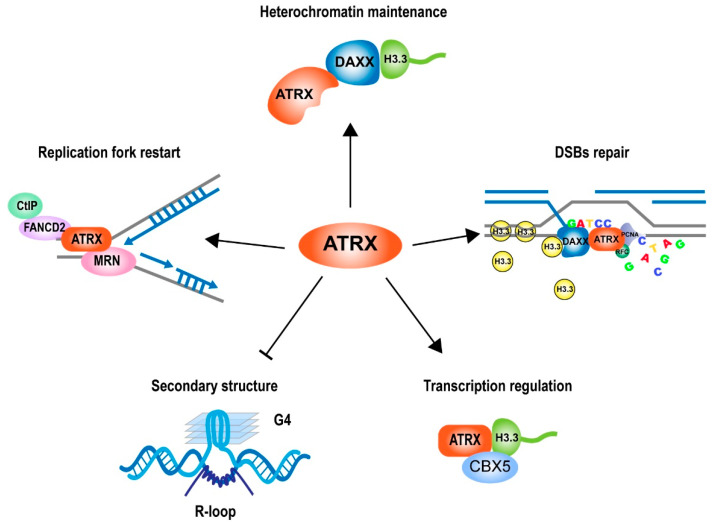
The multiple roles of ATRX in the cellular pathways. ATRX maintains the chromatin state at heterochromatic regions, responds to replication stress and promotes replication fork restart, promotes the DNA damage response (DDR), regulates transcription, and avoids G4 DNA structure. Abbreviations: DSBs, double-strand breaks; G4, G-quadruplex.

**Table 1 cancers-15-02228-t001:** Type of ATRX mutations and their functional consequences.

Type of ATRX Mutations	Functional Consequences
Mutations in the PHD finger (PHDmut)	Reduce enrichment of PHDmut protein to ATRX targets; Reduce PRC2 binding at polycomb targets; Associated with ATRX syndrome;
Mutations in the helicase domains (K1584R)	K1584R accumulates at ATRX targets; Loss of PRC2 binding at some sites and gains at others; Associated with ATRX syndrome;
Point mutations	Result in protein dysfunction and are associated with tumorigenesis

**Table 2 cancers-15-02228-t002:** Molecular marker in diffuse gliomas.

Marker	Biological Function	Diagnostic Methods	Clinical Significance
IDH1 R132 or IDH2 R172 mutation	Gain of function mutation causing gCIMP	Immunohistochemistry for IDH1 R132H followed by sequencing for noncanonical IDH1 or IDH2 mutations	Diagnostic marker for IDH-mutant diffuse gliomas
ATRX mutation/ATRX loss of nuclear expression	Causes alternative lengthening of telomeres	Immunohistochemistry for loss of nuclear ATRX expression or sequencing	Diagnostic marker for IDH-mutant astrocytomas
1p/19q codeletion	Unclear, possibly biallelic inactivation of tumor suppressors on 1p (e.g., FUBP1) or 19q (e.g., CIC)	PCR-based loss of heterozygosity analysis, FISH, array-based copy number analysis, MLPA	Diagnostic marker for IDH-mutant and 1p/19q-codeleted oligodendrogliomas
H3 K27M mutation	Histone 3 mutation causing epigenetic alterations affecting gene expression	Immunohistochemistry for H3 K27M or sequencing	Diagnostic marker for diffuse midline glioma, H3 K27M-mutant

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
