# Peer review of "The Chromatin Remodeler ATRX: Role and Mechanism in Biology and Cancer"

_cancers, 2023, doi:10.3390/cancers15082228_

Round 1

Reviewer 1 Report

Overall this is a well written and comprehensive review paper describing the role and mechanism of ATRX in biology and cancer. 

There are a few changes that should be made before publication:

1. Generally, I think there needs to be more focus on the results of ATRX mutations on the human biology and disease. This is well described from a molecular level, but more should be discussed from a biological, physiological and clinical perspective.

2. Ensure that proteins are all capitalized and genes are both capitalized and italicized. A few examples of this are on lines 77, 78, 251, 441, 447 ATRX should be italicized as you are describing the gene; lines 179 Rad51 should be RAD51 as it is a protein; line 195 Tip60 is a protein so should be TIP60; lines 302, 30, 317, 320 should italicize IDH1 as a gene; italicize genes in line 420

3. Ensure acronyms are defined in their first use. For example, line 83 DAXX, line 173 and 176 HR has already been defined in 161

4. It would be helpful to add a table to the paragraph starting at line 66 to show the different ATRX mutations and their functional consequences.

5. The paragraph starting at line 97 should be placed under 3.2

6. Ensure the tense of the sentences is consistent. Change "resulted" in line 135-136 to "results".

7. Change "in" to "at" in line 141

8. Line  213-214 does not make sense

9. Add a figure that describes ATRX's role in cancer and co-mutations (lines 243-248

10. Better explain the diagnostic algorithm described in lines 246-248.

11. Figure 5 is confusing and hard to read

12. Line 335 MYCH should be MYCN

13. Define functional and non-functional in lines 397-398

Author Response

Response to Reviewer 1

  1. Generally, I think there needs to be more focus on the results of ATRX mutations on the human biology and disease. This is well described from a molecular level, but more should be discussed from a biological, physiological and clinical perspective.

Thanks for your careful review and valuable comments. We agree with the reviewer that more focus on biological, physiological, and clinical perspectives of ATRX mutations should be paid. Please kindly refer to our revised description. 

  1. Ensure that proteins are all capitalized and genes are both capitalized and italicized. A few examples of this are on lines 77, 78, 251, 441, 447 ATRX should be italicized as you are describing the gene; lines 179 Rad51 should be RAD51 as it is a protein; line 195 Tip60 is a protein so should be TIP60; lines 302, 30, 317, 320 should italicize IDH1 as a gene; italicize genes in line 420

Thanks for your careful review. We have revised the manuscript and made the proteins capitalized and genes both capitalized and italicized.

  1. Ensure acronyms are defined in their first use. For example, line 83 DAXX, line 173 and 176 HR has already been defined in 161

Thanks for your valuable suggestion. We have added the acronyms in their first use.

  1. It would be helpful to add a table to the paragraph starting at line 66 to show the different ATRX mutations and their functional consequences.

We are grateful for this valuable suggestion. We have added a table below the paragraph to show the different ATRX mutations and their influences. Please kindly refer to the new table 1.

  1. The paragraph starting at line 97 should be placed under 3.2

This comment is appreciated. We have moved the paragraph under 3.2.

  1. Ensure the tense of the sentences is consistent. Change "resulted" in line 135-136 to "results".

Thanks for your careful review. We have changed the spelling mistake.

  1. Change "in" to "at" in line 141

 Thanks for your careful review. We have changed "in" to "at" in line 141.

  1. Line 213-214 does not make sense

To follow the valuable comment, we have modified the sentence. Please kindly refer to our revised version in lines 203-205.

  1. Add a figure that describes ATRX's role in cancer and co-mutations (lines 243-248

We are grateful for this valuable suggestion. We have added a figure to describe ATRX’s role. Please kindly refer to the revised version in line 256.

  1. Better explain the diagnostic algorithm described in lines 246-248.

We are grateful for this valuable suggestion. We have added a table below the paragraph to explain the ATRX mutations in diagnosis. Please kindly refer to the new table 2 in line 287.

  1. Figure 5 is confusing and hard to read

Thanks for the valuable suggestion. We have redrawn the figure to make it easy to read (line 323, figure 6)

  1. Line 335 MYCH should be MYCN

Thanks for the careful review. We have changed the spelling mistake.

  1. Define functional and non-functional in lines 397-398

We are grateful for this valuable suggestion. We have added the definition of functional and non-functional PanNETs in lines 460-463.

Reviewer 2 Report

This review shows an intact and progressive logical frame that introduce molecular functions and structures first, then discuss its critical interactions in four critical processes, and finally predict the mechanism associated with tumorigenesis according to the cellular processes.

However, a controversy is the contexts related to the concept of “ALT”. Authors introduced the word in a front paragraph of the abstract and descripted it in the background part of ‘mechanism of ATRX in cancers. These details will provide readers an information that you will link the ‘ALT’ to ALRX mutations for discussing the cancer mechanisms. This information also had been mentioned in conclusion. However, look through the whole paragraph of the ‘mechanism of ATRX in cancers’, authors focus more on cellular process like DNA damage, replication stress and methylation but very few details for ‘ALT’ except in first introduction. Therefore, it would be better if you associate this DNA alteration to exact telomere alteration, or just weaken the description of ‘ALT’ in abstract and conclusion. 

Author Response

Response to Reviewer 2

This review shows an intact and progressive logical frame that introduce molecular functions and structures first, then discuss its critical interactions in four critical processes, and finally predict the mechanism associated with tumorigenesis according to the cellular processes.

However, a controversy is the contexts related to the concept of “ALT”. Authors introduced the word in a front paragraph of the abstract and descripted it in the background part of ‘mechanism of ATRX in cancers. These details will provide readers an information that you will link the ‘ALT’ to ALRX mutations for discussing the cancer mechanisms. This information also had been mentioned in conclusion. However, look through the whole paragraph of the ‘mechanism of ATRX in cancers’, authors focus more on cellular process like DNA damage, replication stress and methylation but very few details for ‘ALT’ except in first introduction. Therefore, it would be better if you associate this DNA alteration to exact telomere alteration, or just weaken the description of ‘ALT’ in abstract and conclusion. 

Thanks for the valuable comments. We agree that weakening the description of ‘ALT’make more sense for the work. We have revised the description in the abstract and conclusion. Please kindly refer to the revised version.

Round 2

Reviewer 2 Report

The authors adressed my questions and I have no other concerns.